Regulation of cellular and molecular markers of epithelial-mesenchymal transition by Brazilin in breast cancer cells

Cayetano-Salazar Lorena
Hernandez-Moreno Jose A.
Bello-Martinez Jorge
http://orcid.org/0000-0003-1813-9062 Olea-Flores Monserrat
http://orcid.org/0000-0002-0669-4408 Castañeda-Saucedo Eduardo
http://orcid.org/0000-0001-5386-8530 Ramirez Monica
http://orcid.org/0000-0002-6247-8590 Mendoza-Catalán Miguel A.
http://orcid.org/0000-0003-4911-0545 Navarro-Tito Napoleon nnavarro@uagro.mx
Facultad de Ciencias Químico Biológicas, Autonomous University of Guerrero , Chilpancingo, Guerrero , Mexico
Qin Jiangjiang
Electronic publication date: 2024 May 9
Publication date: 2024
Volume: 12
Electronic Location ID: e17360
Received 2024 Jan 12; Accepted 2024 Apr 18
Copyright: © 2024 Cayetano-Salazar et al.
Copyright year: 2024
Copyright holder: Cayetano-Salazar et al.
License: This is an open access article distributed under the terms of the Creative Commons Attribution License, which permits unrestricted use, distribution, reproduction and adaptation in any medium and for any purpose provided that it is properly attributed. For attribution, the original author(s), title, publication source (PeerJ) and either DOI or URL of the article must be cited.
License URL: https://creativecommons.org/licenses/by/4.0/

Keywords: Breast cancer, Cell invasion, Vimentin, E-cadherin, MMP-9, MMP-2, Brazilin

Funding: COCYTIEG SEP-PROMEP/103.5/14/11118 UAGro- PTC 053 SEP-CONACYT CB-2014-01-239870 This work was supported by grants from COCYTIEG awarded to Jorge Bello-Martinez and SEP-PROMEP/103.5/14/11118 (UAGro- PTC 053) and SEP-CONACYT CB-2014-01-239870 awarded to Napoleon Navarro-Tito. The funders had no role in study design, data collection and analysis, decision to publish, or preparation of the manuscript.

==============================
Breast cancer is the most common invasive neoplasm and the leading cause of cancer death in women worldwide. The main cause of mortality in cancer patients is invasion and metastasis, where the epithelial-mesenchymal transition (EMT) is a crucial player in these processes. Pharmacological therapy has plants as its primary source, including isoflavonoids. Brazilin is an isoflavonoid isolated from Haematoxilum brasiletto that has shown antiproliferative activity in several cancer cell lines. In this study, we evaluated the effect of Brazilin on canonical markers of EMT such as E-cadherin, vimentin, Twist, and matrix metalloproteases (MMPs). By Western blot, we evaluated E-cadherin, vimentin, and Twist expression and the subcellular localization by immunofluorescence. Using gelatin zymography, we determined the levels of secretion of MMPs. We used Transwell chambers coated with matrigel to determine the in vitro invasion of breast cancer cells treated with Brazilin. Interestingly, our results show that Brazilin increases 50% in E-cadherin expression and decreases 50% in vimentin and Twist expression, MMPs, and cell invasion in triple-negative breast cancer (TNBC) MDA-MB-231 and to a lesser extend in MCF7 ER+ breast cancer cells. Together, these findings position Brazilin as a new molecule with great potential for use as complementary or alternative treatment in breast cancer therapy in the future.

Introduction

Worldwide, breast cancer is the most frequent malignant neoplasm in women and is 70–80% curable in early stages in patients with non-metastatic disease (Harbeck et al., 2019). Cell invasion and metastasis are considered the most critical markers of cancer progression as they determine the malignancy of cancer (Guan, 2015; Meirson, Gil-Henn & Samson, 2019). Cell invasion is the first step in the metastatic cascade and is associated with the epithelial-mesenchymal transition (EMT) program (Hanahan, 2022). During EMT, there is loss of cell-cell junctions and downregulation of epithelial markers such as cytokeratins, claudins, occludins, and E-cadherin and increase of mesenchymal markers, such as fibronectin, N-cadherin, ZEB-1, Snail, Slug, Twist and vimentin (Ribatti, Tamma & Annese, 2020). EMT can be regulated by ERK1/2 signaling pathways, tyrosine kinases, transforming growth factor β (TGFβ), insulin-like growth factor (IGF), epidermal growth factor (EGF), platelet-derived growth factor (PDGF), NF-κB, protein kinase B (Akt) and the (Wnt)/β-catenin signaling pathway (Chen et al., 2013; Lindsey & Langhans, 2014; Luo, 2017; Xu, Yang & Lu, 2015). As a consequence, these signaling pathways can activate the transcription factors Twist and Snail, whose primary function is to repress the expression of E-cadherin, a marker of epithelial cells, and increase the expression of vimentin, a marker of mesenchymal cells (Wang, Li & Chen, 2018).

In particular, E-cadherin is a protein that maintains cell-cell junctions and epithelial polarity, whose loss of expression in cancer is associated with the initiation of EMT, invasion and metastasis (Hui San & Ching Ngai, 2024). While vimentin, a type III intermediate filament, provides structural and functional support to cells, protects cancer cells from mechanical stress during migration and provide increased plasticity and is considered as a canonical marker of EMT in cancer (Usman et al., 2021). On the other hand, Twist is a transcription factor considered a master regulator of EMT-related genes, including cell migration, survival, multiple chemoresistance, apoptosis, and immune survival (Chuang, Chiou & Hsu, 2023). Increased expression of Twist is associated with invasion and metastasis and is responsible for repressing E-cadherin expression and promoting the expression of fibronectin, N-cadherin and vimentin, thereby reducing adhesion and increasing cell migration, invasion and metastasis (Farahzadi et al., 2023; Georgakopoulos-Soares et al., 2020). In addition, tumor cells increase secretion of matrix metalloproteinases (MMPs), whose function is proteolytic degradation of basement membrane and extracellular matrix (ECM) proteins (Falk et al., 2018; Gonzalez-Avila et al., 2019). Increased secretion of matrix metalloproteinase-2 (MMP-2) and matrix metalloproteinase-9 (MMP-9) has been reported in invasive and metastatic breast cancer (Jiang & Li, 2021; Li et al., 2017a). Tumor cells that undergo EMT produce increased MMPs, facilitating invasion and metastasis. Therefore, looking for new treatment alternatives in which phytochemical compounds have promising potential, such as isoflavonoids, is vital.

Isoflavonoids are polyphenolic metabolites found abundantly in plants of the Fabaceae family (Veitch, 2013), and have shown antitumor potential inducing cytotoxicity and apoptosis, decrease tumor cell migration, invasion, and metastasis, and have also been shown to control EMT regulatory molecules (Cayetano-Salazar et al., 2021; Uifălean et al., 2015). Isoflavonoids regulate EMT by increasing epithelial marker expression and decreasing mesenchymal marker expression (Barnes, 2010; Li et al., 2009). Previous studies report that the isoflavonoid biochanin A decreased cell migration and invasion in A427 lung cancer cells, by decreased Snail expression and increased E-cadherin expression (Wang, Li & Chen, 2018). On the other hand, combined treatment with doxorubicin and formononetin reversed the EMT process in U87MG glioma cells by decreasing vimentin expression and increasing E-cadherin expression (Quan et al., 2015). The mode of action of isoflavonoids is by blocking TGF-β/IRS, EGF/EGFR, FAK/Paxillin, MAPK, IκB, PI3K/Akt ERK, STAT3, FAK, and Src-mediated signaling pathways (Javed et al., 2021; Li et al., 2011; Pejčić et al., 2023).

Brazilin is an isoflavonoid identified and isolated from Haematoxylum brasiletto and Caesalpinia sappan, which has shown a cytotoxic effect, inhibiting proliferation, and inducing apoptosis (Bello-Martínez et al., 2017; Lee et al., 2013; Nava-Tapia et al., 2022). Brazilin inhibits the nuclear translocation of NF-kB and activation of mTOR, STAT1, and STAT3, inhibiting MAPK signaling (Cayetano-Salazar et al., 2021; Jia, Tong & Fan, 2019; Li et al., 2017b). On the other hand, it promotes the activation of Casp-8, Casp-10, p21, and p27, and the expression of proapoptotic proteins Bax and Bak, and also inhibits the expression of anti-apoptotic proteins Bcl-2 and Bcl-XL (Suyatmi et al., 2022).

The recognition of molecular markers involved in EMT and invasion is fundamental to finding targeted therapies against invasive and metastatic breast cancer; the effect of Brazilin on EMT markers is currently unknown. For this reason, we decided to evaluate whether Brazilin influence the expression of EMT markers such as E-cadherin, vimentin, and Twist, the secretion levels of MMP-2 and MMP-9 and the invasion of MCF7 and MDA-MB-231 breast cancer cells.

Materials and Methods

Brazilin was obtained from H. brasiletto, as previously described. The ethanol extract of dried heartwood of H. brasiletto (1,500 g) was obtained by maceration with EtOH at room temperature for 7 days with regular agitation. The total extract was produced by evaporating the combined ethanol extracts under reduced pressure. The Brazilin compound was purified using flash column chromatography (flash CC), with EtOAc: CHCl3: MeOH gradient elution, resulting in the isolation of Brazilin (0.6 g). They confirm the nature of the compound, a spectroscopic analysis was carried out by 1H and 13C nuclear magnetic resonance (NMR), the sample was dissolved in CDCl3-d. The 1H and 13C NMR spectra were recorded on a Bruker AVANCE III HD instrument. (500 MHz) using TMS as internal reference. The chemical shifts (δ) were determined in parts per million and the values of the coupling constants (J) in hertz (Hernández-Moreno et al., 2023).

FITC conjugated phalloidin was purchased from Cytoskeleton (Denver, CO, USA). Mouse anti-E-cadherin, anti-vimentin, and anti-Twist antibodies were purchased from Santa Cruz Biotechnology (Santa Cruz, CA, USA). Rabbit anti-GAPDH antibody was purchased from ABclonal (Woburn, MA, USA). HRP-conjugated secondary antibodies were obtained from Millipore (Billerica, MA, USA), and Alexa Fluor 488-conjugated anti-mouse antibody was obtained from Invitrogen (Carlsbad, CA, USA).

Cell culture

MCF7 and MDA-MB-231 breast cancer cells were obtained from ATCC (Manassas, VA) and cultured in Dulbecco’s Modified Eagle Medium F12 (DMEM/F12) (50:50, V: V; Sigma-Aldrich, St Louis, MO, USA) with 5% fetal bovine serum (FBS) and 1% antibiotic (penicillin G/streptomycin; Gibco, Waltham, MA, USA). Cells were cultured at 37 °C in a humidified atmosphere with 5% CO2. The MCF7 and MDA-MB-231 breast cancer cells were serum-deprived for 24 h before Brazilin treatment (Olea-Flores et al., 2019).

Brazilin treatment

MCF7, and MDA-MB-231 cells were cultured to 80% confluence in 60 mm dishes in 3 ml DMEM/F12 supplemented with 5% SFB for each experimental condition. The Brazilin treatments were applied at 0, 2.5, 5, 10, 20, and 40 µM in serum 1% medium for 24 h. After this time, the culture medium was removed, and cells were lysed with a Triton X-100 lysis buffer (Tris-HCl, NaCl 1.5 M, EDTA 1.0 mM/pH 8, EGTA 1.0 mM, 1% Triton X-100 and 10% glycerol). The chemical inhibitors of proteases and phosphatases were added: sodium fluoride (NaF) 100 mM, phenylmethylsulfonyl fluoride (PMSF) 1 mM, and sodium orthovanadate (Na3VO4) 1 mM.

F-Actin staining

MCF7 and MDA-MB-231 cells were seeded on glass coverslips, grown to 70% confluence, and stimulated either with or without Brazilin 0, 20 y 40 µM for 24 h. Cells were fixed for 5 min with 4% paraformaldehyde in PBS and permeabilized with 0.2% Triton-X 100 in PBS at room temperature (RT). For F-actin staining, cells were incubated with FITC-conjugated phalloidin (1:1,000 dilution) for 30 min at RT and imaged with an Olympus BX43 microscope, using the 40X objective. Images were analyzed with ImageJ software, version 1.44p (NIH, Bethesda, MD, USA) (Olea-Flores et al., 2019).

Western blot

MCF7 and MDA-MB-231 cell lysates (25 µg) were resolved on 10% SDS-polyacrylamide gels and transferred to nitrocellulose membranes (Bio-Rad, Hercules, CA, USA). To block non-specific binding, the membranes were incubated in 5% (w/v) low-fat milk powder in TBS/T (0.05% Tween 20 in TBS) for 2 h at RT. The membranes were then incubated with the primary antibodies, anti-E-cadherin, anti-vimentin, and anti-Twist overnight at 4 °C (1:1,000 dilution). The membranes were washed with TBS/T and incubated with HRP-conjugated secondary antibodies (1:5,000 dilution) for 2 h at room temperature. The membranes were analyzed using an enhanced chemiluminescence detection system from Bio-Rad (Hercules, CA, USA). Bands obtained were quantified by densitometric analysis using ImageJ software, version 1.44p (NIH, Bethesda, MD, USA) (Olea-Flores et al., 2019).

Immunofluorescence

MCF7 and MDA-MB-231 cells were seeded on glass coverslips, grown to 70% confluence, and treated with Brazilin 0, 20, and 40 µM for 24 h. Cells were then fixed for 5 min with 4% paraformaldehyde in PBS and permeabilized with 0.2% Triton-X 100 in PBS. Cells were then fixed for 5 min with 4% paraformaldehyde in PBS and permeabilized with 0.2% Triton-X 100 in PBS at room temperature. Non-specific staining was blocked with 3% bovine serum albumin in PBS for 2 h at room temperature. Cells were then incubated for 2 h at room temperature with anti-E-cadherin, anti-vimentin, and anti-Twist antibodies (1:200 dilution), followed by a 2-h incubation at room temperature with an Alexa Fluor 488-conjugated anti-mouse secondary antibody (1:200 dilution). Cells were counterstained with 4’6-diamidino-2-phenylindole (DAPI), mounted with Fluoroshield/DAPI media (Sigma-Aldrich). As negative controls, cell cultures were incubated with secondary antibody without the primary antibody to discard basal florescence of the cells. Images were analyzed with ImageJ software, version 1.44p (NIH, Bethesda, MD, USA) (Juárez-Cruz et al., 2019).

Zymography

The culture medium was collected from MCF7 and MDA-MB-231 cells after Brazilin treatments, and proteins were concentrated using 30 kDa filter ultracentrifuge units (Amicon, Merck-Millipore, Burlington, MA, USA). A total of 25 µg of concentrated supernatant from each condition was assayed for proteolytic activity on gels with a gelatin substrate (Olea-Flores et al., 2019). Samples were mixed with non-reducing buffer containing 2.5% SDS, 1% sucrose, and 4 mg/ml phenol red and separated on 8% acrylamide gels polymerized with 1 mg/ml bovine gelatin. Electrophoresis was performed at 72 V for 4 h (Juárez-Cruz et al., 2019). Subsequently, the gels were washed three times with 2.5% Triton X-100 and then incubated in 50 mM Tris-HCl pH 7.4 and 5 mM CaCl2 at 37 °C for 24 h. The gels were stained with 2.5% Triton X-100 for 24 h. The gels were stained with 0.25% Coomassie Brilliant Blue G-250. Excess dye was removed by washing with 10% acetic acid and 30% methanol. Proteolytic activity was detected as clear bands against the background staining of the undigested substrate in the gel (Juárez-Cruz et al., 2019). Quantification was performed using ImageJ software, version 1.44p (NIH, Bethesda, MD, USA).

Cell invasion assays

Matrigel invasion assays were performed following the Transwell chamber method (Kramer et al., 2013), using 24-well plates containing 8 μm pore size inserts (Corning, Kennebunk, ME, USA). 30 µL of Matrigel (Corning Kennebunk, ME, USA) was added to the inserts and kept at 37 °C for 30 min to form a semi-solid matrix (Juárez-Cruz et al., 2019). MDA-MB-231 and MCF7 cells were treated for 2 h with cytosine β-D-arabinofuranoside (AraC) 10 μM to inhibit cell proliferation during the experiment. Then Brazilin treatment was applied at 0 and 20 µM for 48 h. 1 × 105 cells per insert were seeded in a serum-free medium in the upper chamber. 600 µL of DMEM supplemented with 0.1% SFB was added to the bottom of the insert. The cells were incubated for 48 h at 37 °C in a 5% CO2 atmosphere. After this time, the cells and Matrigel were gently removed from the top of the Transwell membrane with a cotton swab. Invading cells were washed and fixed with methanol for 5 min and stained with 0.1% crystal violet diluted in PBS. Cell quantification was performed using a hemocytometer and an Olympus BX43 microscope with a 10X objective. The number of invaded cells were quantified using ImageJ software, version 1.44p (NIH, Bethesda, MD, USA) (Juárez-Cruz et al., 2019).

Statistical analysis

The results are presented as the mean ± SEM. Data were analyzed statistically by one-way ANOVA, and comparisons were performed by Dunnett’s multiple comparison test using the Prism 9.5.1 GraphPad software. A statistical probability of p < 0.05 was considered significant.

Results

Brazilin induces morphological changes in MDA-MB-231 and MCF7 breast cancer cells

Epithelial-mesenchymal transition (EMT) involves a morphologic change from an epithelial to a mesenchymal phenotype where epithelial cells become detached from their neighboring cells and the underlying basement membrane and become more motile and migratory (Olea-Flores et al., 2019). EMT is characterized by loss of epithelial morphology, increased membrane projections, and increased capacity for cell migration and invasion (Santamaria et al., 2017). MDA-MB-231 cells are characterized by highly invasive and mesenchymal morphology. We found by brightfield microscopy that treatment with Brazilin 20 and 40 µM at 24 h induced changes in the number and cell morphology. The morphological changes in MDA-MB-231 cells were observed to be smaller, with a rounded morphology, and with more expansive spaces between cells and a more significant number of detached cells compared to control cells (Fig. 1A). We also observed that Brazilin treatment induces the change to circular morphology in a concentration-dependent manner in MDA-MB-231 cells (Fig. 1B). Similarly, in MCF7 cells with epithelial morphology at 20 and 40 µM Brazilin, cells with a circular phenotype were observed respect to the control (Fig. 1A). However, only at 40 µM significant morphological changes towards a circular phenotype were observed in MCF7 cells (Fig. 1B).

Figure 1 Brazilin induces morphological changes in MDA-MB-231, and MCF7 cells.

(A) Brazilin 10, 20, and 40 µM for 24 h induced morphological changes of MDA-MB-231 and MCF7 cells. (C) Graph of rounded cells of Brazilin treatment to the control. (C) Brazilin at 20 and 40 µM induces changes in actin polymerization in MDA-MB-231 and MCF7 cells. (D) Graphical representation of the effect of Brazilin on morphological changes in MDA-MB-231 and MCF7 cells. Effect of Brazilin on the radius size of MDA-MB-231 (E) and MCF7 (F) cells. Categorization of MDA-MB-231 (G) and MCF7 (H) cells to size: 1–2; 2–4; >4. Differences were determined with one-way ANOVA and Dunnett’s multiple comparison test. Statistical significance: **p < 0.01, ***p < 0.001.

It has been described that the actin cytoskeleton plays an important role in EMT and cancer progression, since, during migration, invasion and metastasis processes, tumor cells generate membrane protrusions driven by actin filaments (Izdebska et al., 2020). In this regard, we evaluated Brazilin-induced morphological changes by staining actin filaments of MDA-MB-231 and MCF7 cells with Phalloidin. We found that Brazilin induces morphological changes and actin filaments formation. Interestingly, Brazilin 20 and 40 µM induces smaller actin filaments and fewer stress fibers respect to unstimulated cells as well as a more epithelial morphology, being more evident in MDA-MB-231 cells (Fig. 1C).

We also performed a graphical representation of the effect of Brazilin on the morphological changes it induces in MDA-MB-231 and MCF7 cells by taking the radius, where the transverse and longitudinal relationship is considered (Fig. 1D). Subsequently, we measured cell radius to determine the size of Brazilin-treated cells relative to the control and found that Brazilin treatment has a significant effect on decreasing the size of MDA-MB-231 (Fig. 1E) and MCF7 (Fig. 1F) cells. When categorizing cell size after Brazilin treatment, the number of cells with circular phenotype and smaller size increased with respect to the control being more evident in MDA-MB-231 cells (Figs.1G and1H).

Brazilin regulates the expression and distribution of E-cadherin, vimentin, and Twist in MDA-MB-231 and MCF7 breast cancer cells

EMT is a mechanism that promotes tumor dissemination and is marked by loss of E-cadherin expression, inhibition of cell adhesion, and induction of cell motility and invasion (Chao, Shepard & Wells, 2010). It has been described that isoflavonoids can regulate the EMT process and, as a consequence, decrease the invasive potential of tumor cells by promoting the expression of epithelial phenotype markers such as E-cadherin and reducing the expression of mesenchymal markers such as N-cadherin, Snail, Slug, and Twist (Hsiao, Ho & Pan, 2020; Liskova et al., 2020). We also evaluated the effect of Brazilin on the expression and distribution of E-cadherin, vimentin, and Twist. We found that in MDA-MB-231 cells, treatment with 10 and 40 µM Brazilin significantly increased the expression of E-cadherin (Figs. 2A and 2B).

Figure 2 Brazilin regulates E-cadherin, vimentin, and Twist protein levels in MDA-MB-231 cells.

Cells were treated with Brazilin 0–40 µM for 24 h. (A) Representative Western blot of the effect of Brazilin on EMT protein levels. Densitometric and statistical analysis of E-cadherin (B), vimentin (C), and Twist (D) levels. GAPDH was used as a loading control. Values represent protein levels relative to control (0), mean ± SEM of three independent experiments were plotted. Differences from the control were determined with one-way ANOVA and Dunnett’s multiple comparison test. Statistical significance: *p < 0.05, **p < 0.01. Representative images of vimentin (E) and Twist (F) by immunofluorescence assays, blue shows nuclei staining. Images were obtained at 40X objective. (G) Model of the effect of Brazilin on the morphological changes of MDA-MB-231 cells. Statistical analysis of vimentin (H) and Twist (I) fluorescence intensity.

Vimentin protein is a canonical EMT marker in mesenchymal cells and is involved in cancer progression (Kidd, Shumaker & Ridge, 2014). Vimentin inhibits focal adhesion-associated proteins to promote cell migration as they confer cells with increased plasticity (Liu et al., 2015; Schaedel et al., 2021). By Western blot assays, we show that Brazilin 10, 20, and 40 µM significantly decreased vimentin expression (Figs. 2A and 2C). While the results of immunofluorescence assays show that MDA-MB-231 cells basally have a high expression of vimentin and, after treatment with Brazilin, decreased its expression and induced a redistribution at the nuclear level showing a more significant effect at 40 µM (Figs. 2E and 2H). In addition, Brazilin induced morphological changes, from the characteristic mesenchymal phenotype to a circular phenotype, with smaller cells than the control (Figs. 2E and 2G).

Twist is a key transcription factor for EMT induction, which promotes cell migration, invasion, and cancer metastasis; it also confers stem-like characteristics to cancer cells and provides chemoresistance (Cao et al., 2018). Twist is also highly expressed in MDA-MB-231 cells; it inhibits E-cadherin expression and increases the expression of mesenchymal markers such as vimentin, fibronectin, α-SMA, and N-cadherin (Serrano-Gomez, Maziveyi & Alahari, 2016). Here, we found that in MDA-MB-231 cells, treatment with brazilin 40 µM significantly decreased Twist expression compared to the control (p < 0.05) (Figs. 2A and 2D). Our immunofluorescence results showed that Twist localizes within the nucleus and that at higher Brazilin concentration, Twist expression is lower, and at 40 µM it exhibited a significant decrease (Figs. 2F and 2I).

About 70% of breast cancers are estrogen receptor-positive, and MCF7 breast cancer cells are an ideal estrogen receptor-positive breast cancer model (Patel et al., 2023). Therefore, we decided to evaluate the effect of the isoflavonoid Brazilin on E-cadherin and vimentin expression in estrogen receptor-positive MCF7 cells. By Western blot, we found that MCF7 cells treated with Brazilin at 24 h presented no changes in E-cadherin expression (Figs. 3A and 3B), whereas, at 20 and 40 µM it decreased vimentin expression (p < 0.001) and presented greater effect concerning triple-negative breast tumor cells (Figs. 2A and 2C).

Figure 3 Brazilin regulates E-cadherin,vimentin, and Twist protein expression levels in MCF7 cells.

The cell cultures were treated with Brazilin 0, 2.5, 5, 10, 20, and 40 µM for 24 h. (A) Representative Western blot of Brazilin effect. Densitometric and statistical analysis of E-cadherin (B), vimentin (C), and Twist (D) levels. GAPDH was used as a loading control. Values represent protein levels relative to control (0), mean ± SEM of three independent experiments were plotted. Differences from the control were determined with one-way ANOVA and Dunnett’s multiple comparison test. Statistical significance: *p < 0.05, **p < 0.01. Representative images of E-cadherin (E) and vimentin (F) immunofluorescence assay, blue shows staining of nuclei. Images were obtained at 40X objective. (G) Model of the effect of Brazilin on the morphological changes of MCF7 cells. Statistical analysis of the fluorescence intensity of E-cadherin (H) and vimentin (I).

In MCF7 cells E-cadherin protein is highly expressed and maintains cell-cell junctions. By immunofluorescence assays we show that treatment for 24 h with Brazilin 20 and 40 µM maintains E-cadherin-mediated cell-cell junctions and Brazilin also increased expression of E-cadherin location at the cell membrane (Fig. 3E). Interestingly, we found that treatment with Brazilin 20 and 40 µM at 24 h decreased vimentin expression (Fig. 3F). Brazilin 40 µM induced morphological changes towards a circular phenotype (Fig. 3G). Fluorescence quantification showed no significant changes in E-cadherin expression (Fig. 3H); however, a significant decrease in vimentin fluorescence intensity was observed (Fig. 3I).

Therefore, we evaluated the effect of Brazilin on Twist expression in MCF7 cells of the epithelial phenotype. The results of the Western blot assay show that Brazilin presented no significant changes in Twist expression in MCF7 breast cancer cells (Figs. 3A and 3D).

Brazilin decreases MMP-9 and MMP-2 secretion and invasion of MDA-MB-231 and MCF7 cells

MMPs are essential during invasion and metastasis; they degrade extracellular matrix (ECM) and basement membrane components (Shay, Lynch & Fingleton, 2015). It has been reported that during breast cancer invasion and metastasis, the secretion and activation of MMP-2 and MMP-9 increase (Stankovic et al., 2010). For this reason, we decided to evaluate the effect of Brazilin on MMP-2 and MMP-9 secretion in MDA-MB-231 cells and MCF7 cells. Cells were treated at different concentrations of Brazilin; we found that Brazilin 20 and 40 µM for 24 h significantly decreased the levels of MMP-2 and MMP-9 secretion in MDA-MB-231 cells (Figs. 4A–4C).

Figure 4 Brazilin decreases MMP-2 and MMP-9 secretion and invasion of MDA-MB-231 cells.

(A) Zymography assays of MDA-MB-231 cells treated with Brazilin 0, 2.5, 5, 10, 10, 20, and 40 µM for 24 h, corresponding to MMP-9 (92 kDa) and MMP-2 (72 kDa) degradation bands. GAPDH was used as a loading control. Densitometric and statistical analysis of MMP-9 (B) and MMP-2 (C) secretion levels in MDA-MB-231 cells. (D) Representative brightfield microscopy images of invasion assays of MDA-MB-231 cells treated with 0, the positive control (medium supplemented with 1% SFB, and Brazilin 20 mM for 24 h. (E) Statistical analysis of the invasion assay of MDA-MB-231 cells. Graphs represent mean ± SEM of three independent experiments. Differences were determined with one-way ANOVA and Dunnett’s multiple comparison test. Statistical significance: *p < 0.05, **p < 0.01.

Currently, invasion and metastasis represent a clinical challenge in the late stages; it is incurable and account for 90% of cancer deaths (Dillekås, Rogers & Straume, 2019; Wang, Zhang & Wang, 2021). To determine whether the decrease in MMP-2 and MMP-9 secretion could be related to the anti-invasive effect, we performed a transwell invasion assay evaluating the concentration of 20 µM Brazilin on the invasion of MDA-MB-231 and MCF7 cells. We used a medium supplemented with 1% SFB as a positive control. We found Brazilin 20 μM significantly decreased the invasive potential of MDA-MB-231 cells (Figs. 4D and 4E).

Finally, in MCF7 cells, Brazilin treatment at 20 and 40 µM significantly decreased MMP-9 secretion (Figs. 5A and 5B), and from 10, 20, and 40 µM decreased MMP-2 secretion (Figs. 5A and 5C). In addition, at 20 µM, cell invasion was decreased (Figs. 5D and 5E).

Figure 5 Brazilin decreases MMP-2 and MMP-9 secretion and invasion of MCF7 cells.

(A) Zymography assays of MCF7 cells treated with Brazilin 0, 2.5, 5, 10, 20, and 40 µM for 24 h, corresponding to degradation bands of MMP-9 (92 kDa) and MMP-2 (72 kDa). GAPDH was used as a loading control. Densitometric and statistical analysis of MMP-9 (B) and MMP-2 (C) secretion levels in MCF7 cells. (D) Representative brightfield microscopy images of invasion assays of MCF7 cells, the positive control (medium supplemented with 1% SFB), and 20 µM Brazilin for 24 h. (E) Statistical analysis of MCF7 cell invasion assays. Graphs represent mean ± SEM of three independent experiments are plotted. Differences were determined with one-way ANOVA and Dunnett’s multiple comparison test. Statistical significance: *p < 0.05, **p < 0.01, ***p < 0.001.

Discussion

Currently, 90% of breast cancer deaths are associated with metastasis, and unfortunately, there is no effective therapy against invasive or metastatic breast cancer (Dillekås, Rogers & Straume, 2019; Meirson, Gil-Henn & Samson, 2019). EMT has been shown to play a central role in cancer invasion and metastasis, where cells undergo a process of dedifferentiation from a “cuboidal” epithelial phenotype to a mesenchymal phenotype, and morphological changes occur as larger, membrane-projecting, highly invasive cells (Olea-Flores et al., 2020). However, evidence suggests that phytochemicals such as isoflavonoids have anticancer effects, as they regulate ROS production, promote cell cycle arrest, induce apoptosis, and inhibit proliferation and invasion of tumor cells (Cayetano-Salazar et al., 2021; Kopustinskiene et al., 2020). Interestingly, Brazilin has been reported to decrease proliferation, induce apoptosis and cell cycle arrest, and reduce migration of breast tumor cells (Nava-Tapia et al., 2022). In this regard, we decided to evaluate the effect of Brazilin on the EMT process, including morphological changes, E-cadherin, vimentin and Twist regulation, MMP-2, and MMP-9 secretion levels, and invasion in breast cancer tumor cells. In our study, we found that Brazilin promoted E-cadherin expression, decreased actin polymerization, vimentin expression, MMPs secretion and cell invasion of MCF7 and MDA-MB-231 breast cancer cells.

Particularly, we found that treatment with Brazilin 20 and 40 µM induced morphological changes in MDA-MB-231 and MCF7 cells, where cells acquire a circular morphology and smaller size. Previously, it has been reported that Brazilin in human glioblastoma U87 cells induced morphological alterations such as decreased cell size, shrinkage, and reduced cell number. They also report that Brazilin induced cell cycle arrest in the Sub-G1 phase and apoptosis by cleavage of caspase-3, caspase-7, and PARP (Lee et al., 2013). On the other hand, in multiple myeloma cells, treatment with Brazilin at 60 µM at 24 h induced cell cycle arrest in the G2/M phase and apoptosis, the morphological features that reported decreased cell size and shrinkage (Kim et al., 2012). Therefore, the round morphology induced by Brazilin in MCF7 and MDA-MB-231 cells could suggest apoptotic cells.

During the EMT process, tumor cells acquire migratory capacity due to invadopodia formation, lamellipodia and filopodia. Changes in tumor cell morphology depend on reorganization of actin cytoskeleton (Haynes et al., 2011). Our results show that Brazilin inhibits actin filament formation in MDA-MB-231 and MCF7 cells. Interestingly, at the highest concentration, fewer membrane projections and smaller cells can be seen, acquiring a round morphology towards a more epithelial phenotype.

During the EMT process, there is a decrease in the expression of E-cadherin and an increase in the expression of vimentin, and the transcription factors Snail and Twist (Loh et al., 2019). E-cadherin maintains cell-cell junctions; it has been established that loss of E-cadherin is involved in the EMT process (Felipe Lima et al., 2016). While the transcription factors Snail and Twist repress E-cadherin expression and increased the expression of vimentin intermediate filament protein (Tian et al., 2020). By Western blot and immunofluorescence, we found that Brazilin decreased the expression of vimentin and Twist in MDA-MB-231 cells. In contrast, in MCF7 cells, it decreased vimentin expression levels. Interestingly, there is evidence that isoflavonoids can reverse the EMT process by decreasing the expression of vimentin, Snail, and Twist and consequently reducing the invasiveness of tumor cells (Cayetano-Salazar et al., 2021; Imran et al., 2019). Other studies have reported that the isoflavonoids genistein (Chan et al., 2018), biochanin, puerarin (Zhou et al., 2020), and glabridin (Hsu et al., 2011) regulate the invasive process of tumor cells by increasing the expression of E-cadherin and decreasing the expression of vimentin, Snail, and Twist. Our results suggest that Brazilin may play a key role in regulating the EMT markers of MDA-MB-231 breast tumor cells.

Another group of molecules whose deregulation is associated with EMT and cell invasion are MMPs; their function consists of the breakdown of extracellular matrix and basement membrane components (Cabral-Pacheco et al., 2020; Scheau et al., 2019). In breast cancer, increased secretion and activation of MMP-2 and MMP-9 promote tumor cell invasion into other tissues and metastasis to distant organs (Jiang & Li, 2021). Our results revealed that Brazilin decreased MMP-2 and MMP-9 secretion in MDA-MB-231 and MCF7 cells. The effect of Brazilin on the inhibition of MMP-2 and MMP-9 secretion could be related to the inhibition of NFkB activation, as suggested by Kim (2010), who report decreased 12-O-tetradecanoylphorbol-13-acetate (TPA)-induced MMP-9 expression and invasion in MCF7 cells (Kim, 2010). Similar effects on MMPs inhibition via NF-kB-induced human dermal fibroblasts have been reported (Lee et al., 2012). We suggest that Brazilin might regulate EMT by decreasing MMP-2 and MMP-9 secretion by inhibiting NF-kB nuclear translocation.

A consequence of the activation of the EMT process in tumor cells is the acquisition of invasive and metastatic properties, due to the inhibition of E-cadherin expression and increased expression of mesenchymal markers such as vimentin and Twist (Sun et al., 2020). It has also been described that overexpression of MMP-2 and MMP-9 is associated with poor prognosis in breast cancer patients by promoting the invasive process due to their role in the degradation of ECM proteins (Jiang & Li, 2021). Furthermore, we found that Brazilin treatment inhibited the invasive potential of MDA-MB-231 and MCF7 cells, which is related to the effect Brazilin exhibited on the upregulation of EMT markers and inhibition of MMP-2 and MMP-9 secretion. It has been previously described that Brazilin inhibits NF-kB activation through interaction with its active site, inhibiting cell migration and invasion (Haryanti et al., 2022). In a recent study, they also report that Brazilin inhibited the migration and invasion of MDA-MB-23 and 4T1 breast cancer cells; however, they do not describe the mechanism by which this occurs (Yang et al., 2023).

Previous studies have reported that Brazilin inhibits NFkB/p50 signaling in cartilage and chondrocytes obtained from osteoarthritis patients (Weinmann et al., 2018). On the other hand, Brazilin inhibited nuclear translocation of the p65 subunit of NFkB in in vivo and in vitro models (Li et al., 2017b). Furthermore, in vascular smooth muscle cells, Brazilin inhibited the phosphorylation and activation of PDGF-Rβ, Src, ERK1/2 and Akt signaling pathways (Guo et al., 2013). Additionally, it has been reported that Brazilin decreased ERK and NFkB signaling in RANKL-stimulated RAW264.7 cells (Kim et al., 2015). While in human umbilical vein endothelial cells, Brazilin suppresses the vascular inflammatory process induced by high glucose, an effect related to inhibition in NF-κB activation (Jayakumar et al., 2014). These results suggest that in breast cancer cells, Brazilin regulates EMT by inhibiting MAPK, PI3K and STAT3 signaling and consequently prevents Twist activation (Fig. 6).

Figure 6 Mechanism of action of Brazilin in regulating EMT markers in MCF7 and MDA-MB-231 breast cancer cells.

Inhibition of MAPK, PI3K, and STAT signaling pathways by Brazilin in breast cancer cells. Brazilin structure was created using the molsoft software (https://www.molsoft.com/mprop/). Image was created using Biorender software (https://www.biorender.com/).

Clinically in cancer, E-cadherin expression has been described to be associated with decreased invasion, growth inhibition, apoptosis, cell cycle arrest and cell differentiation (Wong et al., 2018). In addition, decreased expression of vimentin, Twist, MMP-2 and MMP-9 is associated with inhibition of tumor growth, cell migration, invasion and cancer metastasis and increased sensitivity to chemotherapeutic agents (Jiang & Li, 2021; Satelli & Li, 2011; Winter et al., 2021). Previous reports have described that Brazilin might have lower cytotoxic effects and have specific effects on tumor cells, as observed with other natural compounds (Kopustinskiene et al., 2020; Shin et al., 2020).

However, one of the limitations of this study is that the experiments were performed in vitro using breast cancer cell lines. In this regard, it is necessary to evaluate the effect of Brazilin in animal models and eventually to evaluate its effect in patients in clinical trials. Nevertheless, the results of this study are promising and we suggest that Brazilin has the potential to be used in clinical therapy against TNBC and luminal A breast cancer, either as an alternative or complementary to current therapies in breast cancer.

Conclusions

In this study we found that Brazilin induces morphological changes from a mesenchymal genotype towards a more epithelial or circular phenotype in MDA-MB-231 and MCF7 cells, being more evident in MDA-MB-231 cells, these morphological changes are related to the effect of Brazilin on the regulation of actin polymerization. We found that Brazilin significantly affects the upregulation of EMT markers vimentin and Twist and the invasion of highly invasive MDA-MB-231 breast tumor cells of the TNBC subtype. However, in ER-positive MCF7 cells, Brazilin showed a low effect on the up-regulation of E-cadherin, vimentin, and Twist and cell invasion. Therefore, we suggest that in MCF7 cells, higher concentrations of Brazilin are needed to regulate the EMT process since no changes in E-cadherin and Twist expression were observed. We suggest that Brazilin regulates the EMT process by inhibiting the activation of the transcription factor Twist or indirectly by inhibiting upstream STAT3, PI3K, and ERK1/2 signaling pathways, which regulate Twist activation; however, further studies are needed to test this hypothesis.

Supplemental Information

Supplemental Information 1 Full-resolution micrographs fro. Figure 1.

Supplemental Information 2 Full-resolution micrographs fro. Figure 2.

Supplemental Information 3 Full-resolution micrographs fro. Figure 3.

Supplemental Information 4 Full-resolution of images of the three replicates of gelatin zymography assays.

Supplemental Information 5 Full-resolution micrographs fro. Figure 5.

Supplemental Information 6 Images of the three replicates of the Western blot assays in MDA-MB-231 cells.

Independent replicates of the effect of Brazilin on E-cadherin, vimentin and Twist levels.

Supplemental Information 7 Images of the three replicates of Western blot assays in MCF7 cells.

Independent replicates of the effect of Brazilin on E-cadherin, vimentin and Twist levels.

Supplemental Information 8 Images of the three replicates of gelatin zymography assays.

Independent replicates of the effect of Brazilin on MMP-2 and MMP-9 secretion in MDA-MB-231 and MCF7 cells.

Supplemental Information 9 Raw data of the analyses performed in the experiments of our manuscript.

Densitometric and statistical analysis of the values obtained by western blot and gelatin zymography.

Supplemental Information 10 The photographs of uncorked blots and gels from gelatin zymography.

compendium of compressed images of western blots and gelatin zymography by triplicate.

Additional Information and Declarations

Competing Interests

Author Contributions

Data Availability

The authors declare that they have no competing interests.

Lorena Cayetano-Salazar conceived and designed the experiments, performed the experiments, analyzed the data, prepared figures and/or tables, and approved the final draft.

Jose A. Hernandez-Moreno conceived and designed the experiments, performed the experiments, analyzed the data, prepared figures and/or tables, and approved the final draft.

Jorge Bello-Martinez conceived and designed the experiments, analyzed the data, prepared figures and/or tables, and approved the final draft.

Monserrat Olea-Flores conceived and designed the experiments, performed the experiments, authored or reviewed drafts of the article, and approved the final draft.

Eduardo Castañeda-Saucedo conceived and designed the experiments, authored or reviewed drafts of the article, and approved the final draft.

Monica Ramirez conceived and designed the experiments, performed the experiments, authored or reviewed drafts of the article, and approved the final draft.

Miguel A. Mendoza-Catalán conceived and designed the experiments, performed the experiments, authored or reviewed drafts of the article, and approved the final draft.

Napoleon Navarro-Tito conceived and designed the experiments, analyzed the data, prepared figures and/or tables, and approved the final draft.

The following information was supplied regarding data availability:

The raw data are available in the Supplemental File.

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
