# Peer review of "Regulation of cellular and molecular markers of epithelial-mesenchymal transition by Brazilin in breast cancer cells"

_PeerJ, doi:10.7717/peerj.17360_

## Round 0.1 · original submission · Major Revisions

Please carefully read the comments and suggestions from the reviewers and provide your point-point responses.

Reviewer 1 ·

Basic reporting

No comments.

Experimental design

Methods described with sufficient detail & information to replicate.

Validity of the findings

Conclusions are well stated, linked to original research question & limited to supporting results.

Additional comments

Comments:
I appreciate the efforts that have been made by the authors. There are a few pieces of feedback that I believe will contribute to making the manuscript clearer
Abstract
1. The abstract is well-written. However, I recommend introducing the full terms of MMP initially and subsequently using the abbreviation.
2. Mentioning the specific type of breast cancer investigated would strengthen the abstract. you would need to specify the type of breast cancer cells you are using in your experiments (IDC, ILC, TNBC, HER2+, IBC).
3. Instead of "promotes" and "decreases", state the actual percent change observed in protein levels or other measurements. For example, Western blot analysis showed a [quantify]% increase in E-cadherin and a [quantify]% decrease in vimentin expression.
4. Replace "canonical markers" with specific examples like "E-cadherin, vimentin, and Twist". For example, this study investigated the effect of brazilin, an isoflavonoid isolated from Haematoxilum brasiletto with known antiproliferative activity, on key EMT markers like E-cadherin, vimentin, and Twist expression.
5. It would be nice to specify the cancer cell line that was used in your experiment.
6. Highlight the novelty and potential advantages of brazilin compared to existing therapies.
Introduction
The introduction provides a good foundation for your research, but it could be strengthened by focusing on the most relevant information and improving clarity and flow.
1. The introduction is quite lengthy. Consider trimming unnecessary details or phrases to improve readability. For example, the sentences about specific genes regulated by isoflavonoids can be condensed.
2. While providing background information is important, prioritize information directly relevant to your study (brazilin, EMT markers, invasion).
3. The transition between sections could be smoother. Briefly summarize key points and connect them to your research question to enhance logical flow.
4. Avoid redundant information (e.g., mentioning EMT twice in consecutive sentences).
5. Start by directly stating the problem of breast cancer metastasis and highlighting the role of EMT.
6. Briefly describe EMT and its key markers (E-cadherin, vimentin, Twist).
7. Introduce isoflavonoids and their potential to inhibit EMT and metastasis.
8. Briefly mention brazilin and its known anti-cancer effects.
9. Clearly state your research question: Does brazilin influence EMT markers and invasion in breast cancer cells?
Methods
The method section clearly stated the experiments. Some minor comments can make the flow of this section, better.
1. Define abbreviations at their first use (e.g., DMEM/F12, FBS, PBS).
2. Simplify sentences where possible (e.g., "Cells were maintained in a humidified atmosphere with 5% CO2 at 37 °C" can be condensed to "Cells were cultured at 37 °C in a humidified atmosphere with 5% CO2").
3. Describe the software used in the statistical analyses.
4. By specifying the source and type of both positive and negative controls in your immunofluorescence method section, you provide important information the validity of your conclusions.
5. By providing the information regarding the criteria used for quantifying protein bands and cell invasion, you allow readers to understand how you analysed your data and ensure the robustness of your results.

Results
This section is well-written, and the figures are clear. A few minor comments could enhance the clarity of this section.
1. Combine subsections "Brazilin regulates the expression levels and distribution of E-cadherin, vimentin, and Twist in..." and "Brazilin decreases MMP-9 and MMP-2 secretion and invasion of..." for each cell line. This avoids repetition and improves flow.
2. Define abbreviations at their first use (e.g., EMT, ECM).
3. Use consistent verb tenses (e.g., use "found" instead of "showed" consistently).
4. Avoid redundancy (e.g., replace "decreased vimentin levels" with "decreased vimentin expression" when the meaning is the same).
Discussion
This part is comprehensive; however, some feedback can improve the organisation and the consistency of it.
1. Combine similar paragraphs. For example, merge the paragraphs discussing brazilin's effects on morphology and actin cytoskeleton, and the paragraphs discussing MMPs and the NF-kB pathway.
2. Briefly summarize key findings at the beginning of the discussion to provide context.
3. Avoid redundancy and unnecessary repetition.
4. Briefly cite evidence instead of fully describing it (e.g., instead of saying "It has been reported that brazilin inhibited UVB-induced MMP-1/3 secretion via inhibition of NF-kB-induced human dermal fibroblasts (Y. R. Lee et al., 2012)," just say "Similar effects on MMP inhibition via NF-kB have been reported (Lee et al., 2012).").
5. Use consistent verb tenses.
6. Mention potential clinical implications of your findings.
7. Acknowledge any limitations of the methods used.

Reviewer 2 ·

Basic reporting

Please see attached comments.

Experimental design

Please see attached comments.

Validity of the findings

Please see attached comments.

Additional comments

This work investigated the regulatory effects of the isoflavone compound brazilin on canonical markers (E-cadherin, vimentin, MMPs) in breast cancer cells in vitro. The manuscript needs to be substantially revised before it can be published. Here are some comments.
1. The author prepared brazilin from plants, but the physical and chemical characterization of the isolated product, including NMR spectroscopy, mass spectrometry, etc., was not provided in the manuscript, and the purity information was also unknown.
2. The effects of brazilin on the morphology of breast cancer cells were investigated. However, the effects of brazilin on cell viability at these concentrations has not been reported. It is important to determine the concentration of toxicity. At toxic concentrations, the changes in cell morphology may be related to cell death, rather than the compounds inhibiting the invasive properties of the cells.
3. Figure 6 shows the mechanism by which brazilin inhibits breast cancer cell invasion as hypothesized by the authors, but the discussion surrounding it is inadequate. The author only stacks relevant studies in the discussion section, which lacks logic.
4. The caption of the Figure should be checked carefully. Such as Figure 2G, Figure 2H, and Figure 2I.
5. Please check the chemical formula in the manuscript carefully! Such as “CHCl3” (Line 101), “CDCL3-d” (Line 103), “CaCl2” (Line 168).

Annotated reviews are not available for download in order to protect the identity of reviewers who chose to remain anonymous.

---

## Round 0.2 · accepted · Accept

The authors have addressed the comments from reviewers.

Reviewer 1 ·

Basic reporting

My concerns have been addressed accurately.

Experimental design

My concerns have been addressed accurately.

Validity of the findings

My concerns have been addressed accurately.

Additional comments

My concerns have been addressed accurately.

Reviewer 2 ·

Basic reporting

The manuscript has been improved. There is no further comment.

Experimental design

The manuscript has been improved. There is no further comment.

Validity of the findings

The manuscript has been improved. There is no further comment.

Additional comments

The manuscript has been improved. There is no further comment.